# Impact of Hydrolyzed Collagen from Defatted Sea Bass Skin on Proliferation and Differentiation of Preosteoblast MC3T3-E1 Cells

**DOI:** 10.3390/foods10071476

**Published:** 2021-06-25

**Authors:** Lalita Chotphruethipong, Thunwa Binlateh, Pilaiwanwadee Hutamekalin, Rotimi E. Aluko, Surapun Tepaamorndech, Bin Zhang, Soottawat Benjakul

**Affiliations:** 1International Center of Excellence in Seafood Science and Innovation, Faculty of Agro-Industry, Prince of Songkla University, Hat Yai, Songkhla 90110, Thailand; lalitac.ch@psu.ac.th; 2School of Geriatric Oral Health, Institute of Dentistry, Suranaree University of Technology, Nakhon Ratchasima 30000, Thailand; thunwa.bin@gmail.com; 3Division of Health and Applied Sciences, Faculty of Science, Prince of Songkla University, Hat Yai, Songkhla 90110, Thailand; pilaiwanwadee.h@psu.ac.th; 4Department of Food and Human Nutritional Sciences, University of Manitoba, Winnipeg, MB R3T 2N2, Canada; 5National Center of Genetic Engineering and Biotechnology Center (BIOTEC), National Science and Technology Development Agency (NSTDA), 113 Thailand Science Park, Pathumthani 12120, Thailand; surapun.tep@biotec.or.th; 6College of Food and Pharmacy, Zhejiang Ocean University, Zhoushan 316022, China; zhanbin@zjou.edu.cn

**Keywords:** bioactive peptides, sea bass, osteoblast, differentiation, hydrolyzed collagen, bone

## Abstract

Osteoporosis is a serious problem affecting health of the elderly. Drugs (bisphosphonates) applied for treatment are often accompanied by adverse side effects. Thus, fish byproduct-derived peptides, particularly hydrolyzed collagen (HC) from defatted sea bass skin, could be a safe source of anti-osteoporosis agents. This study aimed to examine the effects of HC on proliferation and differentiation of preosteoblast cells. HC prepared using papain before Alcalase hydrolysis was determined for molecular weight (MW) distribution. Thereafter, the resulting HC (50–800 µg/mL) was added to the cell. Proliferation, alkaline phosphatase activity (AP-A) and mineralization of cells were investigated. Moreover, the expression of runt-related transcription factor 2 (RUNX2) and the p-Akt/Akt pathway were also determined using Western blot. The results showed that HC had an MW < 3 kDa. HC (50–200 µg/mL) could promote cell proliferation. Nevertheless, HC at 100 µg/mL (HC-100) had enhanced AP-A and increased mineralization during the first 7 days of culture. Moreover, HC-treated cells had higher calcium depositions than the control (*p* < 0.05). Additionally, cells treated with HC-100 had higher levels of RUNX2 and p-Akt expressions than control (*p* < 0.05). Therefore, HC could be a promising functional ingredient to promote osteoblast proliferation and differentiation, which could enhance bone strength.

## 1. Introduction

Osteoporosis is a severe problem found in the elderly, which is mainly associated with bone fragility [1]. Lack of estrogen or nutrients, aging and chronic diseases are the main causes of osteoporosis [2]. Recently, bisphosphonates have been used to prevent or treat osteoporosis [3,4], but their uses are accompanied by negative side effects, especially the risk for gastrointestinal toxicity [5].

To avoid this limitation, the natural compounds with high biological activity and safety such as hydrolyzed collagen (HC) from fish processing byproducts could be ap-plied to alleviate osteoporosis owing to their biocompatibility, profound activity and rapid in vivo clearance [6,7]. Additionally, fish HC could be utilized in many foods without the limitation or unacceptability due to cultural and religious restrictions [8]. In general, collagen is the main constituent of bone extracellular matrix [9]. Collagen serves as a template and may also initiate and propagate mineralization independent of the matrix vesicles, resulting in bone toughness and bone strength [10]. Fish HC has been proven to possess the ability to promote bone formation. Yamada et al. [11] revealed that fish collagen peptides could increase collagen synthesis and mineralization of osteoblastic MC3T3-E1 cells. Moreover, hydroxyproline content, alkaline phosphatase level and mineral density of aged mice were enhanced after ingestion of HC from silver carp skin [7]. Thus, fish HC is promising for inducing differentiation of osteogenic cells, leading to augmented bone formation.

HC from defatted sea bass skin is one of the excellent sources of peptides rich in various bioactivities, including fibroblast proliferation, antioxidant and wound-healing activities [12,13]. Moreover, the sea bass skin contains peptides with Pro-Hyp or X-Hyp-Gly [14], which could promote differentiation of MC3T3-E1 cells [15]. Nevertheless, effects of HC from defatted sea bass skin on the proliferation and differentiation of MC3T3-E1 pre-osteoblast cells have not been reported. Thus, the purpose of the present study was to examine the cell proliferation and differentiation property of the HC from defatted sea bass skin with emphasis on calcium deposition and protein expression.

## 2. Materials and Methods

### 2.1. Chemicals

Alcalase and papain were purchased from Siam Victory Chemicals Co, Ltd. (Bangkok, Thailand). Porcine pancreas lipase (PPL), BCIP/NBT and other chemicals were supplied by Sigma-Aldrich Chemical Co. (St. Louis, MO, USA). Antibiotics, fetal bovine serum (FBS) and modified α-Minimum Essential Medium Eagle (α-MEM) without ascorbic acid were obtained from Gibco^®^, Thermo Fisher Scientific Inc. (Waltham, MA, USA). MC3T3-E1 subclone 4 was procured from ATCC (Bethesda, MD, USA). RUNX2 and Akt, p-Akt (s473) were obtained from Abcam, Cambridge, UK, and Cell Signaling (Danvers, MA, USA), respectively.

### 2.2. Enzyme Assays

PPL activity was measured following the protocol of Chotphruethipong et al. [1]. One unit of activity was defined as the amount of PPL that produces 1 μmol *p*-nitrophenol (*p*-NP)/min. Activities of papain (pH 7.0, 40 °C) and Alcalase (pH 8.0, 50 °C) were assayed for 15 min using casein as substrate. One unit of activity was defined as the amount of papain or Alcalase that liberates 0.01 µmol of tyrosine equivalent/min (µmol Tyr equivalent/min).

### 2.3. Preparation and Pretreatment of Sea Bass Skin Using Pulsed Electric Field (PEF)

The skins (2 × 2 cm^2^) were first treated with 0.1 M NaOH and subsequently washed with tap water [1] prior to PEF pretreatment [2]. Electric field intensity at 24 kV/cm with 72 ms pulses was used. Pulse duration, specific energy input and pulse repetition times were 0.1 ms, 135 kJ/kg and 20 ms, respectively. When PEF treatment was completed, the skins were swollen using 0.05 M citric acid for 2 h, followed by washing until a neutral pH was reached [2].

### 2.4. Defatting of PEF-Treated Skin

Swollen skins (200 g) were mixed with 1 L of PPL solution containing 42.36 U/g dry matter in a vacuum chamber [3]. The vacuum impregnation process was performed for 4 cycles under 100 kPa pressure [3]. The defatted swollen skin was further used for preparing hydrolyzed collagen (HC).

### 2.5. HC Preparation

A 2-step hydrolysis was used for producing the HC. Firstly, the defatted swollen skin was treated with papain (0.3 U/g dry matter) at 40 °C for 90 min. Heating at 90 °C for 15 min was subsequently applied to inactivate papain. The obtained hydrolysate was further hydrolyzed using Alcalase (0.3 U/g dry matter) at 50 °C for 90 min, followed by inactivation of enzyme via heating (90 °C, 15 min). The resulting digest was filtered and lyophilized as the HC powder following the procedure of Chotphruethipong, Aluko and Benjakul [1].

### 2.6. Size Distribution

Molecular weight of HC was determined by MALDI-TOF-MS, as detailed by Benjakul et al. [4].

### 2.7. Impact of HC on Proliferation and Differentiation of MC3T3-E1 Preosteoblast Cells

#### 2.7.1. Cell Culture

MC3T3-E1 cells were cultured in complete α-MEM medium containing 10% FBS and antibiotics, as described by Paisrisarn et al. [5].

#### 2.7.2. Cell Proliferation Assays

HC was dissolved in α-MEM medium containing 10% or 1% FBS to obtain different concentrations (50, 100, 200, 400 and 800 µg/mL). HC at all concentrations was tested for cell proliferation using 3-(4, 5-dimethylthiazol-2-yl)-2, 5-diphenyl tetrazolium bromide (MTT) assay [16]. Moreover, cell morphology after treatment with HC was visualized using a microscope (Olympus IX70 with DP50, Shinjuku-ku, Tokyo, Japan). The levels rendering the highest cell proliferation were selected for measurement of alkaline phosphatase activity and mineralization studies.

Before testing, MC3T3-E1 cells (5 × 10^3^ cells/well) were seeded in a 24-well culture plate using the differentiation medium (DM) containing 10% FBS, 50 µg/mL ascorbic acid, 1 µM dexamethasone and 10 µM β-glycerophosphate followed by incubation for 12 h. Thereafter, the medium was removed and cells were washed with 1 mL of sterile phosphate buffer (SPB). DM with HC at the selected concentrations was added into cells and cultured for 7, 14 and 21 days. The medium was changed every 2–3 days. At the designated days, cells were analyzed for alkaline phosphatase activity and mineralization.

#### 2.7.3. Alkaline Phosphatase Activity (AP-A)

The *p*-nitrophenyl phosphate (*p*-NPP) assay was used for measuring AP-A in cells [6]. The optical density was detected at a wavelength of 405 nm using a microplate reader (FLUOstar Omega, BMG Labtech Multi-mode microplate reader, Offenburg, Germany). AP-A was reported as micromolar (µM) using a standard curve according to the manufacturer’s instructions.

For staining, the levels of AP-A were also assessed using BCIP/NBT alkaline phosphatase liquid substrate. The cells were incubated with BCIP/NBT solution for 1 h in the dark. After staining, the cells were visualized for AP-A levels using a microscope. The blue intensity was analyzed using ImageJ 1.41 software (NIH, Bethesda, MD, USA) and presented as relative intensity to that of control cells.

#### 2.7.4. Determination of Mineralization in MC3T3-E1 Cells

After incubation with HC at the designated time, the cells were washed gently with SPB and fixed with 4% paraformaldehyde for 30 min. After fixing, the cells were stained with 1% alizarin red dye for 1 h at room temperature. Thereafter, the stained cells were washed with SPB and visualized using a microscope. Calcium deposition of cells indicated by alizarin red staining was examined using ImageJ software and was reported as a relative calcium deposition to that of control. Apart from analysis using ImageJ software, mineralization of cells was measured using absorbance at 405 nm. Alizarin red-stained cells were rinsed with 10% cetylpyridinium chloride and incubated for 1 h at room temperature. The resulting color of red mineralized nodules was measured using an absorbance at 405 nm.

Cells treated with HC at concentration and incubation time that produced the highest AP-A and mineralization in cells were used for Western blot analysis. Cells without HC treatment at the same incubation time were also analyzed.

#### 2.7.5. Western Blot Analysis

After treatment with the selected conditions, cells (1 × 10^6^ cells/35-mm dish) were lysed with RIPA buffer containing freshly added protease inhibitor cocktail. The obtained lysates were centrifuged at 32,869× *g* (Beckman Coulter, Inc., Palo Alto, CA, USA) for 15 min at 4 °C. The supernatants were collected and measured for total protein concentration using the Bradford method (Bio-Rad Laboratories, Hercules, CA, USA). The proteins (30 µg) were equally subjected to SDS-PAGE and transferred onto PVDF membranes. Non-specific bindings were blocked with 5% skimmed milk or 3% BSA. The membranes were then incubated with primary antibodies against Akt, p-Akt (s473) and RUNX2, followed by incubation with HRP-conjugated secondary antibody for 1.5 h. The proteins of interest were detected using Super Signal West Pico chemiluminescence substrate (Thermo Fisher Scientific, Rockford, IL, USA) and visualized by film exposure. The density of the protein band was quantified using ImageJ software normalized by β-actin expression (Cell Signaling Technology, Danvers, MA, USA).

#### 2.7.6. Effect of HC Combined with Inhibitor on AP-A and Mineralization in Cells

Cells (5 × 10^3^ cells/35 mm dish) were seeded into a 24-well culture plate. Subsequently, the cells were treated with DM or DM containing HC at the selected concentration alone or pre-treated with 10 µM Akt inhibitor (perifosine) for 1 h prior to the addition of DM containing HC at the selected concentration. After treatments, the cells were incubated for the selected time. At the end of culture period, the cells were determined for AP-A and mineralization using Alizarin red S staining as mentioned above.

### 2.8. Statistical Analysis

The experiments were run in triplicate and analysis of variance (ANOVA) was conducted for all the data. The Duncan’s multiple range test was used for mean comparison. Statistical Package for Social Science (SPSS 11.0 for windows, SPSS Inc., Chicago, IL, USA) was used for data analysis. For Western blot analysis, the difference between samples was analyzed using a paired t-test.

## 3. Results

### 3.1. Size Distribution of HC

Several peaks of peptides having varying molecular weights (MWs) were observed in HC (Figure 1). Peptides with MWs of 888 and 1108 Da were dominant in HC. Moreover, peptides with low MW (<888 Da) and large MW (>1 kDa) were also detected.

### 3.2. Impact of HC on Proliferation, Alkaline Phosphatase Activity and Mineralization of MC3T3-E1 Cells

#### 3.2.1. MC3T3-E1 Cell Proliferation

The effect of HC on proliferation of MC3T3-E1 cells is depicted in Figure 2. In the presence of 10% FBS, HC at all concentrations (50–800 µg/mL) tested had no cytotoxicity on the MC3T3-E1 cells (Figure 2A). Moreover, cell viability was similar between the control (without HC) and those treated with HC at various concentrations (*p* > 0.05). This was ascertained by cell integrity of all samples (Figure 2C). When the level of FBS was reduced to 1%, the proliferation of cells still increased with increasing concentrations of HC up to 200 µg/mL (*p* < 0.05) (Figure 2B). Thus, HC at 200 µg/mL (or lower) was used for further studies.

#### 3.2.2. Alkaline Phosphatase Activity (AP-A)

HC at all concentrations tested enhanced the differentiation of mesenchymal stem cells to osteoblast cells during the early stages, as evidenced by the increased AP-A with increasing incubation times (*p* < 0.05) (Figure 3A). When comparing AP-A between the samples at the same incubation time tested, there was no difference in AP-A of cells treated with HC at 50 and 200 µg/mL within the first 7 days of culture (*p* > 0.05), while HC at 100 µg/mL showed the highest AP-A (*p* < 0.05) (Figure 3A). Nevertheless, HC-treated cells had higher AP-A than the control (without HC) (*p* < 0.05). The results indicate that HC could induce differentiation of osteoblast cell as evidenced by the increased intensity of blue color (Figure 3B,C). After day 7 of culture, no difference in AP-A for all samples tested was found (*p* > 0.05), except on the 14th day for cells treated with 200 µg/mL HC, which had the lowest AP-A (Figure 3A).

#### 3.2.3. Mineralization of MC3T3-E1 Cells

Mineralization of MC3T3-E1 cells was evaluated using calcium staining by Alizarin red S with the absorbance at 405 nm (A_405_), as illustrated in (Figure 4). HC at all tested concentrations increased the cell mineralization as a function of time (*p* < 0.05), as indicated by the increased calcium deposition (red spot) (Figure 4A) and the augmented absorption at 405 nm (Figure 4B). On day 7, all samples tested had no difference in absorption intensity (Figure 4B) and the calcium deposition (Figure 4C) (*p >* 0.05). When the cell culture time was prolonged, higher absorption and increased calcium deposition were found for HC-treated cells than those of control, especially treatments with HC at 100 and 200 µg/mL, which had the drastic increase in A_405_ (*p* < 0.05). For calcium deposition, the highest nodule formation was observed when treated with HC at 100 and 200 µg/mL, compared to the control (Figure 4A,C). However, on day 21, the same trend was found, but the relative deposition of calcium was lower when compared to day 14 (Figure 4C).

### 3.3. Effect of HC on RUNX2 Expression in MC3T3-E1 Cells

Based on the results shown in Figure 3 and Figure 4, cell differentiation was obviously found during the first 14 days of the culture period. Additionally, there was no difference in cell differentiation between 100 and 200 µg/mL HC treatments (*p* > 0.05). Thus, the cells treated with 100 µg/mL HC for 7, 10 and 14 days were selected for measuring the protein levels of RUNX2 using Western blotting. Higher expression of RUNX2 was found in cells treated with HC when compared to the control (*p* < 0.05) for all culture times tested (Figure 5). Moreover, the expression level of RUNX2 in cell treated with HC increased up to the first 10 days of culture period (*p* > 0.05) (Figure 5A,B). Subsequently, the decrease in RUNX2 expression was found at day 14 of the culture period (*p* < 0.05) (Figure 5A,B).

### 3.4. Effect of HC on Akt Signaling Pathway in MC3T3-E1 Cells

As presented in (Figure 6A,B), no difference in the expression level of Akt was found for the control at all times tested, while p-Akt was upregulated with increasing time (Figure 6A). To further investigate whether Akt signaling was related with HC in stimulation of the differentiation process of cells, the effect of HC incorporated with Akt inhibitor (perifosine) on AP-A (Figure 6C,D) and calcium deposition was examined (Figure 6E,F). The lowest AP-A was found for the untreated control cells (*p* < 0.05) (Figure 6C,D). In the presence of HC, AP-A was markedly increased (*p* < 0.05). When HC in combination with perifosine was used, AP-A level was decreased (*p* < 0.05), suggesting inhibition of the Akt signaling pathway. Additionally, there was no difference in absorption value for all samples tested within the first 7 days, regardless of inhibitor or HC addition (*p* > 0.05) (Figure 6F). The results suggest that calcium in cells was produced to a similar extent since cells were still in the early stage.

## 4. Discussion

The MWs of peptides in HC from the present study (Figure 1) were smaller than those of peptides from sea bass skin previously reported by Benjakul et al. [4,7]. In general, the size of peptides is a vital factor affecting bioactivity of HC [8]. Peptides of smaller sizes mostly exhibited higher bioactivities, especially osteoblast proliferation activity [9,10]. In addition, short chain peptides could be rapidly digested and absorbed in the human body [11]. The result implied that sea bass skin HC containing small peptides might have ability to proliferate osteoblast cells.

When HC was added into MC3T3-E1 osteoblast cells (Figure 2), it could promote proliferation and differentiation of MC3T3-E1 osteoblast cells, especially at the concentrations of HC up to 200 µg/mL. The increased proliferation of cells indicated that HC could induce the proliferation of osteoblast cells. Peptides rich in hydrophobic AAs and imino acids might play a vital role in proliferation [9,12]. Moreover, HC rich in Pro-Hyp and Pro-Hyp-Gly have been shown to induce the growth of osteoblast cells [12,13], and a previous work has demonstrated the presence of these peptides in HC from sea bass skin [14]. Apart from peptides containing Pro-Hyp and Pro-Hyp-Gly, low molecular weight peptides (<1 kDa) had the ability to enhance osteoblast proliferation [9,15]. Our previous work showed the presence of low MW peptides (<750 Da) in defatted sea bass skin HC [7], which may have contributed to stimulation of osteoblast proliferation.

When MC3T3-E1 cells were used to determine the impact of HC on cell differentiation, HC could induce differentiation of osteoblast cells, as witnessed by the increased AP-A (Figure 3A). In general, AP is a marker of bone formation and differentiation of osteoblasts [16], which hydrolyzes pyrophosphate and provides inorganic phosphate to promote mineralization in osteoblasts [16]. Thus, the addition of HC peptides with high imino acids or hydrophobic AAs, particularly at the level of 100 µg/mL, more likely promoted differentiation of bone cells during the first 7 days of the culture period. Nevertheless, no change in AP-A was observed after 7 day of culture. This phenomenon might be due to the different stages of cell differentiation. This result was confirmed by differences in the intensity of the blue color on day 14 and day 21 (Figure 3B,C).

Apart from AP-A, HC could also stimulate mineralization of osteoblast cells (Figure 4), which was consistent with the AP-A results (Figure 3). Changes in the differentiation process of osteoblast cells plausibly occurred from the early stage to final stage after 7 days of culture, leading to the increased mineralized nodule formation, as indicated by the augmented absorption (Figure 4B) and enhanced calcium deposition (Figure 4C). The mineralization process is commonly the final step of osteogenic differentiation [17]. During this process, calcium phosphate nanocrystals are produced by osteoblast cells, which are aligned in the collagen based fibrous matrix [18]. The addition of HC from defatted sea bass skin might aid the promotion of cell matrix mineralization, thus leading to enhanced mineralized nodule formation.

When MC3T3-E1 cells were used to examine the effect of HC on the regulation of RUNX2 genes associated with osteoblast differentiation during the early phase, cells treated with HC showed higher expression of RUNX2 than the control (*p* < 0.05) for all culture times tested (Figure 5), indicating accelerated differentiation of MC3T3-E1 cells by the collagen peptides. However, the decrease in RUNX2 expression after 10 days of the culture period (Figure 5A,B) indicated the change in differentiation phase of cells from early phase to last phase, as shown by the increased calcium deposition of the cells (Figure 4C).

Furthermore, Akt signaling, which is a key pathway regulating several processes, including proliferation, differentiation and survival of cells, was investigated [19]. The upregulation of p-Akt with increasing time was obtained (Figure 6A), indicating that differentiation of cells occurred. This might be due to phosphorylation of Akt at Ser473, thus sustaining the phosphorylated/activated levels [20]. Activated Akt commonly stimulates its downstream signaling cascades to promote transcriptional and translational activities of targeted proteins, namely, RUNX2, Osterix [21]. During bone differentiation, activated Akt is required to enhance RUNX2 transcription, increase AP-A and promote bone mineralization [22,23].

Additionally, the incorporation of HC with perifosine (Akt inhibitor) revealed that HC could stimulate cell differentiation in the early stage via the Akt pathway as indicated by the decreased AP levels when perifosine was added (Figure 6C,D). Thus, cell mineralization could not occur, as ascertained by the fewer mineralized nodules with the presence of perifosine (Figure 6E,F).

## 5. Conclusions

HC from defatted sea bass skin had peptides with MW <3 kDa. HC stimulated growth and differentiation of osteogenic MC3T3-E1 cells in the first stage and resulted in acceleration of calcium deposition in the last stage. Addition of HC to the cells led to increased expression of RUNX2 proteins coupled with enhanced calcium deposition, which were attributed to activation of the Akt signaling pathway. Therefore, HC from fish skin can serve as a functional ingredient and nutraceutical for bone strengthening, in which potential alleviation of osteoporosis and related diseases can be achieved. Nevertheless, the efficiency of HC on prevention of osteoporosis needs to be further investigated using bone resorption and osteoclast differentiation both in in vitro and in vivo models.

## Figures and Tables

**Figure 1 foods-10-01476-f001:**
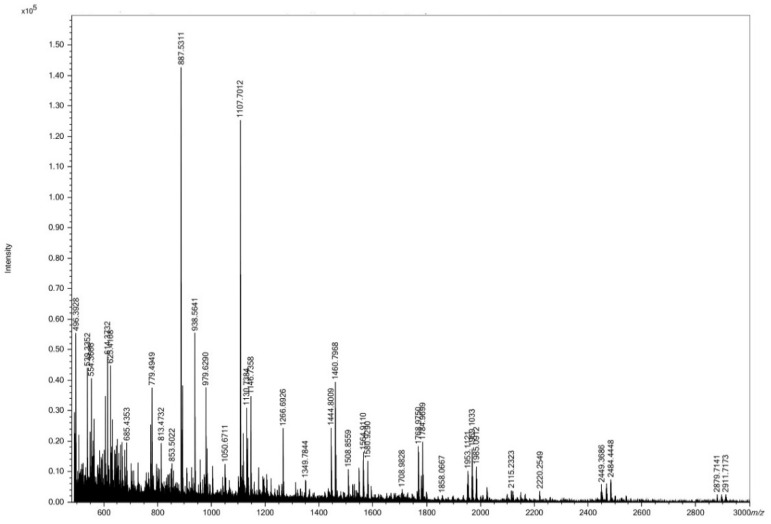
Size distribution of sea bass skin hydrolyzed collagen (HC) determined by MALDI-TOF mass spectrometry.

**Figure 2 foods-10-01476-f002:**
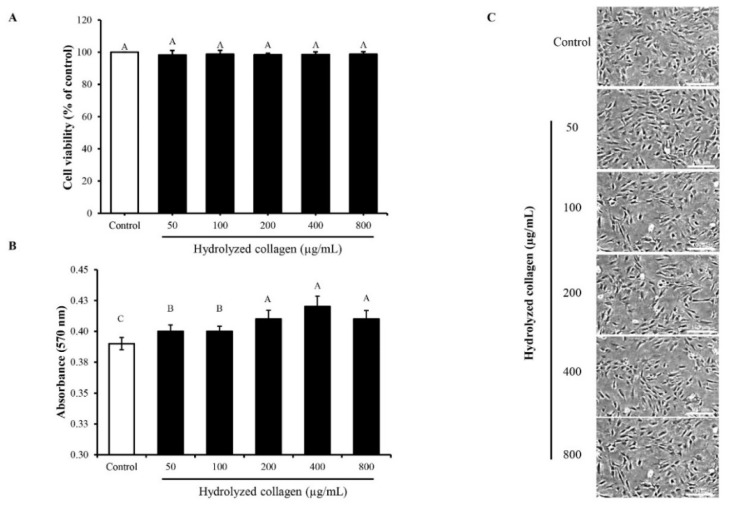
Effect of defatted sea bass skin hydrolyzed collagen (HC) dissolved in α-MEM medium with 10% FBS (**A**) or 1% FBS (**B**) at various concentrations on proliferation of MC3T3-E1 pre-osteoblast cells. Cell morphology using scale bar of 100 μm and magnification of ×10 (**C**) after treatment with HC at various concentrations in the presence of 10% FBS. Incubation time of 24 h was used. Error bars represent standard deviation (*n* = 3). Different letters on bars indicate significant differences (*p* < 0.05).

**Figure 3 foods-10-01476-f003:**
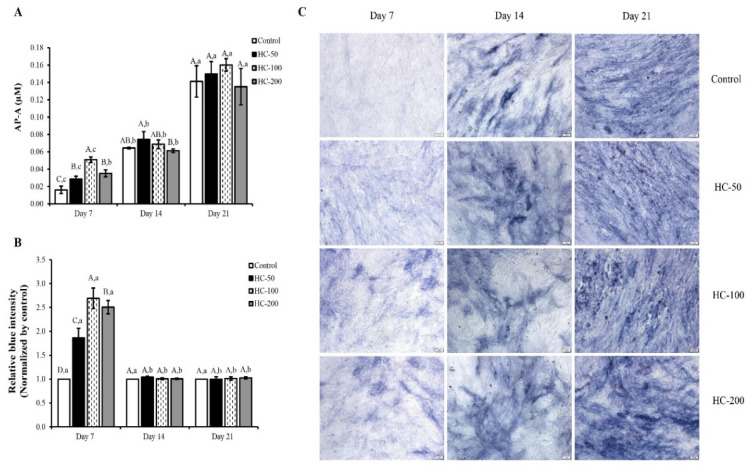
Effect of defatted sea bass skin hydrolyzed collagen (HC) on (**A**) alkaline phosphatase activity (AP-A) of MC3T3-E1 cells, (**B**) relative blue intensity of alkaline phosphatase and (**C**) blue crystal staining intensity (scale bar = 100 μm; magnification = ×10) at different culture times. Quantitative analysis of blue intensity was assessed using ImageJ software. Error bars represent standard deviation (*n* = 3). Different uppercase letters indicate significant differences between samples at the same culture time (*p* < 0.05). Different lowercase letters indicate significant differences between culture times of the same sample (*p* < 0.05).

**Figure 4 foods-10-01476-f004:**
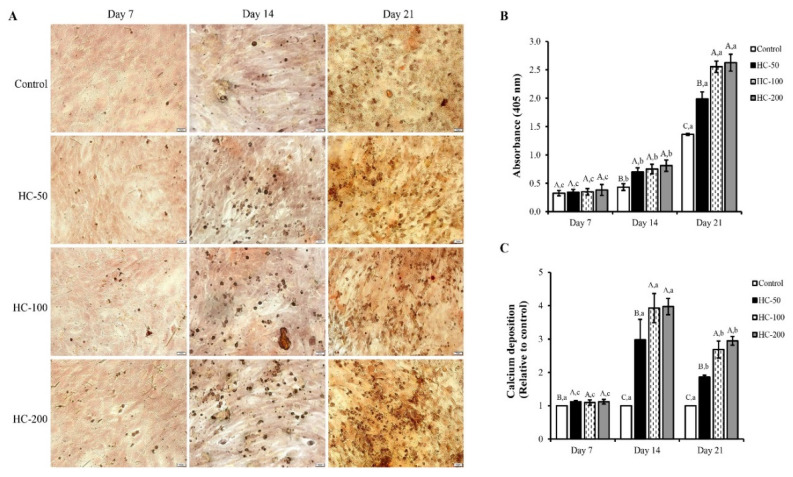
Effect of defatted sea bass skin hydrolyzed collagen (HC) on (**A**) mineralization of MC3T3-E1 cells (scale bar = 100 μm; magnification = ×10) as indicated by (**B**) changes in absorbance at 405 nm and (**C**) calcium deposition during 21 days. Relative calcium deposition was assessed using ImageJ software. Error bars represent standard deviations (*n* = 3). Different uppercase letters indicate significant differences between samples at the same culture time (*p* < 0.05). Different lowercase letters indicate significant differences between culture times of the same sample (*p* < 0.05).

**Figure 5 foods-10-01476-f005:**
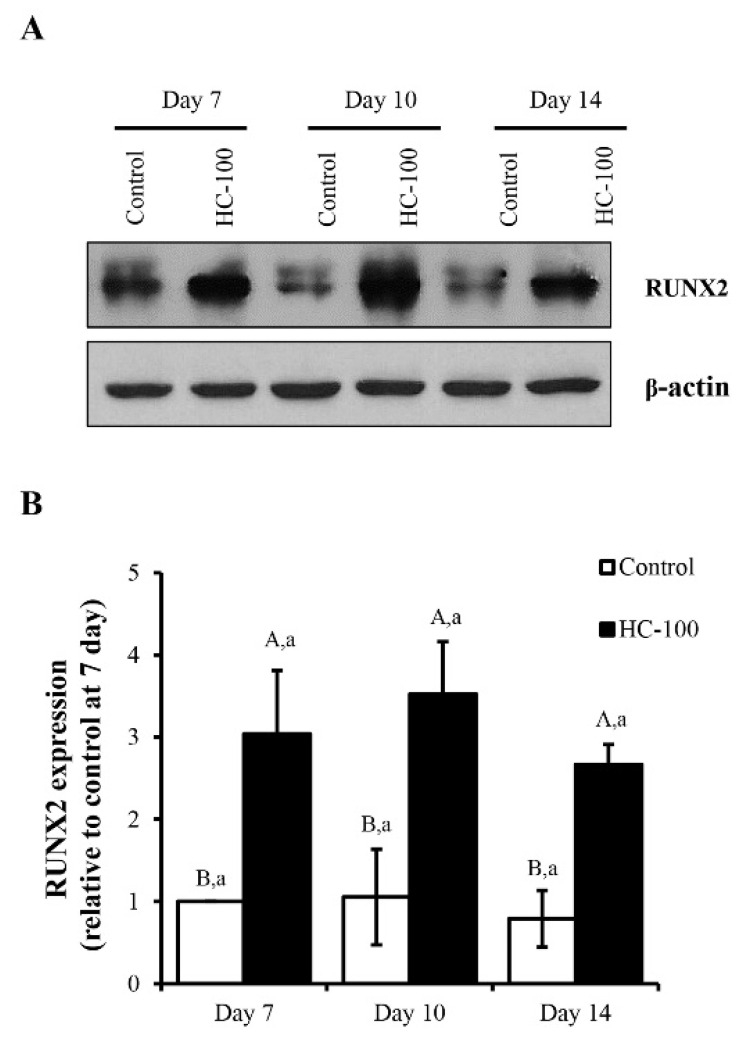
Effect of sea bass skin hydrolyzed collagen (HC) at 100 µg/mL on (**A**) expression level of RUNX2 in MC3T3-E1 cells after differentiation for 7, 10 and 14 days; (**B**) results are expressed as relative values to control at day 7. Error bars represent standard deviation (*n* = 3). Different uppercase letters indicate significant differences between samples at the same culture time (*p* < 0.05). Different lowercase letters indicate significant differences between culture times of the same sample (*p* < 0.05).

**Figure 6 foods-10-01476-f006:**
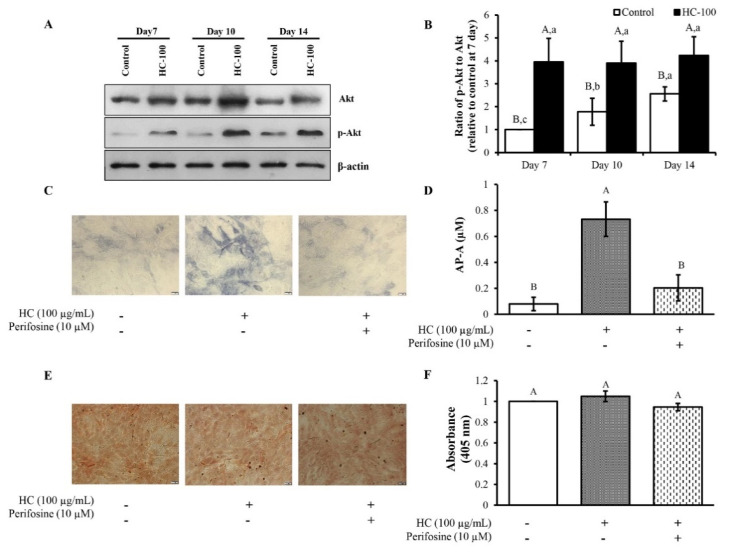
Effect of sea bass skin hydrolyzed collagen (HC) at 100 µg/mL on (**A**) expression of p-Akt, total Akt and β-actin protein expression of MC3T3-E1 cells after differentiation for 7, 10 and 14 days; (**B**) results are expressed as relative value to control at day 7; (**C**) images of blue crystal after staining with BCIP/NBT solution for 1 h; (**D**) alkaline phosphatase activity (AP-A); (**E**) mineralization of cells; and (**F**) changes in absorbance at 405 nm after treatment with or without perifosine (10 μM) for 7 days of differentiation. Error bars represent standard deviation (*n* = 3). Different uppercase letters indicate significant differences between samples at the same culture time (*p* < 0.05). Different lowercase letters indicate significant differences between culture times of the same sample (*p* < 0.05).

## Data Availability

The data are not shared.

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
