# Peer review of "Impact of Hydrolyzed Collagen from Defatted Sea Bass Skin on Proliferation and Differentiation of Preosteoblast MC3T3-E1 Cells"

_foods, 2021, doi:10.3390/foods10071476_

Round 1
Reviewer 1 Report
The authors studied how hydrolyzed collagen from defatted sea bass skin on proliferation and differentiation of preosteoblast MC3T3-E1 cells.
Overall, this is a well-written and well-organized research article.
You should describe more about the significance of this study and future research in the Conclusions.
The scale of the Y-axis should be changed to 0.3-0.4 for an easy reading.
Author Response
The authors studied how hydrolyzed collagen from defatted sea bass skin on proliferation and differentiation of preosteoblast MC3T3-E1 cells. Overall, this is a well-written and well-organized research article.
********Thank you for understanding in our work. All queries have been responded and the corrections have been made and highlighted in yellow.
You should describe more about the significance of this study and future research in the conclusions.
********We have provided the additional information in ‘conclusion’ as per the reviewer’s suggestion. Please see the attachment. Thank you for invaluable comment.
The scale of the Y-axis should be changed to 0.3-0.4 for an easy reading.
********We do agree with reviewer’s comment. Scale of the Y-axis has been changed to 0.3-0.45 for an easy reading. Apart from scale of the Y-axis, ‘line graph’ has been changed to ‘bar graph’ following the third reviewer’s suggestion. Please see Figure 2. Thank you for the insightful comment.

Reviewer 2 Report
The present study is very relevant for the difficult treatment of osteoarthritic problems. It will be the subject of further studies.
Author Response
The present study is very relevant for the difficult treatment of osteoarthritic problems. It will be the subject of further studies.
********In this study, we focused on the effect of hydrolyzed collagen (HC) from defatted sea bass skin on proliferation and differentiation of preosteoblast MC3T3-E1 cells. We found that HC (at 50-200 µg/mL) could promote cell proliferation. Moreover, HC at 100 µg/mL could also enhance alkaline phosphatase activity and increase mineralization of cell in early stage. This reflected that HC could be a promising compound for bone strengthening or alleviation of the diseases related with bone density loss. Regarding the reviewer’s suggestion on the use of HC in preventing osteoarthritic problems, it will be taken to consideration. We plan to conduct the experiment in bone resorption and osteoclast differentiation both in vitro and in vivo models to prove the efficacy of HC in preventing osteoporosis. The results or details for further study will be reported in our next manuscript, which we have provided the information for our future works in conclusion (highlighted in yellow). Thanks so much for the insightful and invaluable comments.

Reviewer 3 Report
The article entitled "Impact of hydrolyzed collagen from defatted sea bass skin on proliferation and differentiation of preosteoblast MC3T3-E1 cells" is interesting and matching to the scope of the journal. However, there are serious issues to address.
It is very difficult to read fonts shows in the figures (axis bars and significant symbols). I recommend to increase quality of figures.
Replace figure 2B with bar chart.
Figure 2c - This panel is too small and it not represent the morphology of cells. Please replace this figure with more clear and magnified image.
Figure 5. The relative intensity calculation method for RUNX2 is wrong. The level of b-actin are not similar. Specifically, B-actin band size of HC-100 after 10 days treatment very small compared to the Day 7 control band. therefore I can not accept this evaluation method as authors are fail to use equal amount of proteins for each tested group. Thus I recommend to replace this figure with bands taken from different trial.
Figure 6- Same issue observed with the B-actin band.
Author Response
The article entitled "Impact of hydrolyzed collagen from defatted sea bass skin on proliferation and differentiation of preosteoblast MC3T3-E1 cells" is interesting and matching to the scope of the journal. However, there are serious issues to address.
********Thank you for invaluable comment and suggestion. All queries have been responded and the corrections have been made following the reviewer’s comments.
It is very difficult to read fonts shows in the figures (axis bars and significant symbols). I recommend to increase quality of figures.
********Sorry for this. The resolution of all figures has been improved for readability. Please see in the revised figures. Thank you.
Replace figure 2B with bar chart.
********Figure 2B has been changed to bar chart as suggested by the reviewer. Thank you.
Figure 2c - This panel is too small and it not represent the morphology of cells. Please replace this figure with more clear and magnified image.
*******We increase the magnification of image as suggested by the reviewer. Please see in Figure 2C Also, scale bar and magnification of image have been included in figure caption (highlighted in green).
Thank you for the comment.
Figure 5. The relative intensity calculation method for RUNX2 is wrong. The level of β-actin are not similar. Specifically, β-actin band size of HC-100 after 10 days treatment very small compared to the Day 7 control band. Therefore, I can not accept this evaluation method as authors are fail to use equal amount of proteins for each tested group. Thus, I recommend to replace this figure with bands taken from different trial.
********Based on alkaline phosphatase activity and cell mineralization results (Figure 3 and Figure 4), the differentiation of cells was obviously found during the first 14 days of culture period. Thus, this period was selected for RUNX2 expression. In our experiment, we monitored the effect of hydrolyzed collagen on differentiation of cells every 7 day for 21 days. Thus, cell without HC treatment (control) cultured using the differentiation medium at day 7 was used for comparing the intensity of protein with the treated samples. We do agree with the reviewer that β-actin (housekeeping gene) is usually used as a loading control for western blot to normalize the levels of protein. Thus, band intensity of protein in all treatments tested should be equal. Before loading protein into gel, we measured the amount of protein using Bradford assay and conducted the trial in triplicate (n = 3). We have to apologize for the mistake in choosing the image. We have changed the image representing actin bands following the reviewer’s suggestion. Please see Figure 5. Thank you for invaluable comment.
Figure 6- Same issue observed with the β-actin band.
********β-actin band of all the samples has been replaced with the new images. Please see Figure 6. Thank you for comment.

Round 2
Reviewer 3 Report
The authors revised this manuscript as expected.